# High Photoresponse Black Phosphorus TFTs Capping with Transparent Hexagonal Boron Nitride

**DOI:** 10.3390/membranes11120952

**Published:** 2021-12-01

**Authors:** Dewu Yue, Ximing Rong, Shun Han, Peijiang Cao, Yuxiang Zeng, Wangying Xu, Ming Fang, Wenjun Liu, Deliang Zhu, Youming Lu

**Affiliations:** 1College of Materials Science and Engineering, Shenzhen University, Shenzhen 518060, China; yuedewu@szu.edu.cn (D.Y.); xmrong@szu.edu.cn (X.R.); Hsdf52690@126.com (S.H.); pjcao@szu.edu.cn (P.C.); zengyx@szu.edu.cn (Y.Z.); wyxu@szu.edu.cn (W.X.); m.fang@szu.edu.cn (M.F.); liuwj@szu.edu.cn (W.L.); dlzhu@szu.edu.cn (D.Z.); 2Key Laboratory of Optoelectronic Devices and Systems of Ministry of Education and Guangdong Province, College of Optoelectronic Engineering, Shenzhen University, Shenzhen 518060, China

**Keywords:** black phosphorus (BP), hexagonal boron nitride (h-BN), thin film transistors (TFTs), n-type, photodetector

## Abstract

Black phosphorus (BP), a single elemental two-dimensional (2D) material with a sizable band gap, meets several critical material requirements in the development of future nanoelectronic applications. This work reports the ambipolar characteristics of few-layer BP, induced using 2D transparent hexagonal boron nitride (h-BN) capping. The 2D h-BN capping have several advantages over conventional Al_2_O_3_ capping in flexible and transparent 2D device applications. The h-BN capping technique was used to achieve an electron mobility in the BP devices of 73 cm^2^V^−1^s^−1^, thereby demonstrating n-type behavior. The ambipolar BP devices exhibited ultrafast photodetector behavior with a very high photoresponsivity of 1980 mA/W over the ultraviolet (UV), visible, and infrared (IR) spectral ranges. The h-BN capping process offers a feasible approach to fabricating n-type behavior BP semiconductors and high photoresponse BP photodetectors.

## 1. Introduction

Two-dimensional crystals have emerged as a class of materials that may impact future electronic technologies, in which, BP, with a puckered structure and a one-atom-thick sheet morphology, was recently discovered by isolating the material from layered BP crystals [1]. In order for phosphorene to be stable and, unlike graphene, to have an inherent, direct, and appreciable band gap, it depends on the number of layers and the in-layer strain and that it is significantly larger than the bulk value of 0.31–036 eV [2]. The direct band gap is 1.7 eV in monophosphorene [3,4]. These properties, together with a remarkably high hole mobility of 1000 cm^2^V^−1^s^−1^ and an on/off ratio of 10^5^ at room temperature [5], make BP suitable for semiconductor applications such as photodetectors [6], solar cells [7], and digital electronics [8]. Recently, a BP single detector spectrometer, enabled by the strong stark effect and the tunable light-matter interactions, showed remarkable potential in the reconstruction of the spectra of both monochromatic and broadband light [9]. The Alberto G. Curto group also demonstrated tunable and stable infrared emission from defect-engineered few-layer BP [10]. However, asymmetries between the hole and electron transport characteristics in BP devices [11,12] complicate the use of BP alone in complementary logic circuits, in which the availability of only p-type semiconductor characterize BP devices. Thus, the need for symmetric charge transport materials has created a serious bottleneck in complementary metal oxide semiconductor (CMOS) application development, in which n-type transistors are needed to reduce static power consumption in logical circuits and systems. Several modulations have been reported, such as Al_2_O_3_ capping [13], depositing Cs_2_CO_3_ and MoO_3_ layers on top of BP [12] in order to achieve the electron- and hole-doping effect. Our group also reported the first air-passivated ambipolar BP transistor by applying benzyl viologen [14]. These methods modulated the surface of BP effectively but obstructed the high photodetection performance of BP TFTs, due to blocking or destructing the surface of BP. Nevertheless, these approaches are not suitable for high photoresponse BP TFT applications because these protective layers render BP limited while n-type BP with high photoresponsivity is required.

The combined method using surface charge transfer doping (HfO_2_ or/and MgO) and passivation (h-BN) was developed, which increases electron mobility [15]. The combined method with BN protection and subsequent thermal annealing provides an effective strategy for a record-long lifetime, with 80% of photoluminescence intensity remaining after 7 months [16]. Bilayer phosphorene FET devices were fabricated of a van der Waals heterostructure with BN, which exhibits tightly bound excitons and trions with two-fold anisotropy [17]. The combined methods can either increase electron mobility or maintain long lifetime photoluminescence, which is unilateral and complex. However, by preventing a decrease in the photoresponse of BP TFTs with n-type behavior, the high transparency property of h-BN becomes apparent, which is also smooth and flat [12,18]. Moreover, 2D h-BN is highly flexible and easy to transfer to other 2D materials using the dry transfer method [19].

In this work, we report a method for converting few-layer BP field-effect transistors (FETs) from p-type to n-type. We report on the achievement of ambipolar characteristics in BP devices by capping with transparent flexible h-BN. The h-BN capping suppressed the degradation of BP upon exposure to ambient conditions, thereby transforming p-type BP to ambipolar BP. Due to transparent h-BN capping, the photoresponsivity of the fabricated BP-based photodetectors increased to 1980 mA/W, much higher than the photoresponsivities reported previously [6]. Our findings demonstrate that BP with transparent h-BN capping offers a robust material for use in transparent high-efficiency optoelectronic and logical electronic applications.

## 2. Materials & Methods

### 2.1. Device Fabrication Process

First, we exfoliated few-layer BP flakes from bulk BP crystals using Scotch tape and transferred them onto a highly p-type-doped silicon substrate capped with a 300 nm SiO_2_ layer under Ar in a glove box. The photoresist polymethyl methacrylate (PMMA) layer was immediately spin-coated onto the samples to protect the flakes from reacting with air. E-beam-lithograph (EBL) patterns were applied to the samples, and Ti/Au layers were deposited (20 nm/50 nm). The samples after the lift-off step are shown in Figure 1a. A transparent h-BN cap was also applied to BP, as shown in Figure 1a. The technique was used to tune the BP FET behavior from p-type to ambipolar.

### 2.2. Electrical Properties and Photoresponse Measurements

The electrical performances of the devices were measured using a probe station under vacuum conditions and a semiconductor parameter analyzer (Agilent 4155C). A xenon arc lamp connected to a monochromator with a mechanical chopper was used as a light source during the photoresponse measurements. The photocurrent was measured using a monochromatic light with wavelength from 200 to 800 nm and an illumination diameter of 60 µm. The power of the light was measured by an optical power meter. For our device, the generated photocurrent can be easily observed when the light is on.

## 3. Results and Discussion

As we know, BP crystals are not stable at ambient conditions [2,5,11,20]. Environmental chemicals, such as H_2_O or O_2_, can easily defect BP, which is also reported by recent theoretical studies [20]. The comparison of different areas on the same exfoliated BP flake was explored; significant roughness develops without protection, while parts of BP under graphene or h-BN do not show any noticeable surface changes, according to the acquired AFM images and as confirmed by the Raman spectroscopy spectrum [13]. The passivated area has clear Ag^1^, B_2g_ and A^2^g peaks as expected, which rules out reactions under air. Thus, capping BP using transparent h-BN effectively improved electron transport in BP devices better than other techniques [12,13,21,22,23] by blocking acceptors derived from the air or environment.

These effects rendered BP suitable for transparent photodetector and logical circuit applications. An optical microscope (OM) image of few-layer BP FET capping with h-BN is shown on the left side of Figure 1a. The right side of Figure 1a shows the Raman spectra of an exfoliated few-layer BP where the BP signature peaks are observed at 364, 438, and 465 cm^−1^, corresponding to the three vibrational modes (A^1^_g_, B_2g_, and A^2^_g_) of the BP crystal lattice [24,25,26], respectively. In addition, the h-BN Raman spectrum peak located at 1370 cm^−1^ corresponding to the vibrational modes (E_2g_) is shown in the inset. In this work, all the Raman spectra were measured after device fabrication, including the chemical treatment steps, the electrical measurements, and the photoresponse tests, to prevent BP devices under fabrication from being damaged by the Raman spectroscopy. Final device structure is depicted in Figure 1b after using polydimethylsiloxane (PDMS) to stack h-BN onto the top of the BP devices.

The output characteristics of the device (hBN1) are shown in Figure 2a. The excellent linear I_ds_–V_ds_ curve suggested good contact between the metal electrodes and the BP flakes. Figure 2b presents the ambipolar transfer characteristics typical of h-BN-capped or pristine p-type BP devices, in which the device performances were initially measured prior to h-BN capping. In contrast with the clear p-type transistor behavior, the red line reveals ambipolar characteristics, in which both electron and hole currents were observed. This transition was attributed to effective acceptor blocking from the air and chemicals. Figure 2d shows the equilibrium state; BP transistors always show p-type behavior. To enhance electron transport, h-BN was used to cap the BP devices and avoid acceptor formation in the presence of air or environmental chemicals, such as H_2_O or O_2_, as shown in the bottom of Figure 2f. Thus, after transparent h-BN capping, the Fermi level was positioned closer to the conduction band than it was prior to treatment, as shown in the top of Figure 2e, indicating an improvement in the electron transportation. Ambipolar characteristics were also obtained from the h-BN-capped BP devices (hBN2, hBN3) in Appendix A.

The Fermi level was dragged toward higher energies to the conduction band, which enhanced electron transport and reduced hole transport. The field effect carrier mobility (μ) was calculated according to the equation:(1)$$\mu =\left(\;1/C_{\mathrm{ox}}\right)\times \left({d\;I}_{\mathrm{ds}}/{d\;V}_{g}\right)\times \left(L_{\mathrm{ch}}/W_{\mathrm{ch}}\right)/V_{\mathrm{ds}}$$
where C_ox_ = 1.15 × 10^–8^ F/cm^2^ for 300 nm SiO_2_, L_ch_ is the channel length, and W_ch_ is the channel width. The electron mobility in the BP devices increased to 73 cm^2^V^−1^s^−1^, a record-high electron mobility, to our knowledge.

A rapid broadband photoresponse was observed among the few-layer BP FETs, and this response was used in ultrafast and wide spectrum response photodetectors [6]. This letter describes the high-performance photodetector properties of ambipolar BP FETs. The photocurrent generated upon monochromatic light illumination (wavelength ranging from 200 to 900 nm) was measured at different V_g__s_ values, revealing clear photocurrent generation as shown in Appendix A. We first measured the photoresponses widely across the UV, visible, and IR wavelength ranges, and we evaluated two important parameters of photodetector performance: the photoresponsivity (R) and the external quantum efficiency (EQE). R and EQE were evaluated by calculating the respective values:(2)$$R=I_{\mathrm{ph}}/P,$$
(3)EQE=hcR/e×100,
where R is defined as the photocurrent generated per power unit of incident light across the effective area, h is Planck’s constant, c is the speed of light, I_ph_ is the photocurrent |I_light_−I_dark_|, P is the power intensity per unit area, and e is the unit charge. Figure 3a shows that R and EQE reached 1980 (mA/W) and 250, respectively. The photoresponsivity of our device was much higher than the values reported previously in BP photodetectors [6], even higher than the values reported in TMDS-based photodetectors [27,28,29,30,31,32]. Our results were comparable to those obtained from certain 2D-based p-n junctions [19,33,34]. The figures of merit of the photodetectors are compared in Table 1. Figure 3a,b show a plot of the photocurrent as a function of the incident power under different gate voltages, indicating the reliability of the device for photocurrent generation. After h-BN capping, Figure 2e shows that the Fermi level was dragged upwards to the conduction band of BP, which narrows the barrier for electron transport in the BP channel. In addition, under the illumination, the photo-generated charge carriers in the BP channel could undergo a thermally assisted tunneling process passing through the barrier to metal, which contributes to the increase of channel current. Together with the tuned Fermi level and the transparent property of h-BN, the photo-induced electrons could penetrate the barrier more easily, producing the enhanced photoresponse for h-BN capping devices, as shown in Figure 3d. That is, due to the transparent property of h-BN and the modified Fermi level from the h-BN capping, photo-induced electrons can easily transport across the barrier. Thus, high performance BP photo-detectors are obtained. In, addition, the time-dependent photoresponse characteristics of the BP photodetectors under different V_ds_ and V_g_ conditions are shown in Figure 3c. The fast photoresponse time of BP is shown in Appendix A, which indicates no degradation in the response time. This apparent photodetector behavior suggests that BP is a promising material for photoelectronic applications.

## 4. Conclusions

We report the fabrication of ambipolar BP devices fabricated through the novel method: capping using transparent h-BN. Our devices exhibited ambipolar characteristics with an electron mobility of 73 cm^2^V^−1^s^−1^. The ambipolar BP devices were used to demonstrate high-performance photodetector behavior. The photoresponsivities in the devices were found to be 1980 mA/W and no degradation in the response time was observed. This photoresponsivity value exceeds the values previously reported. The mechanisms underlying the efficient carrier transport and photoresponsivity were thought to derive from control over the Fermi level in the BP. Our results demonstrated that the method is very effective in enhancing electron transport in the BP transistors, and BP shows promise as a key building block for future 2D semiconductors.

## Figures and Tables

**Figure 1 membranes-11-00952-f001:**
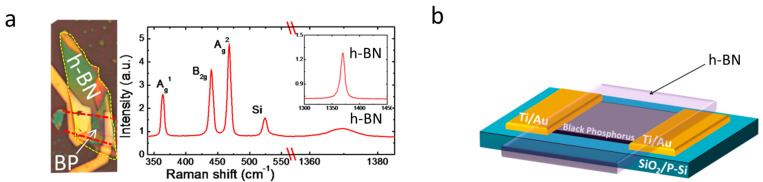
Black phosphorous (BP) devices used in this study. (**a**) OM image of BP FET. Raman spectra of the BP and h-BN used in this study. Inset shows the peak corresponding to h-BN spectra. (**b**) h-BN-encapsulated BP FET.

**Figure 2 membranes-11-00952-f002:**
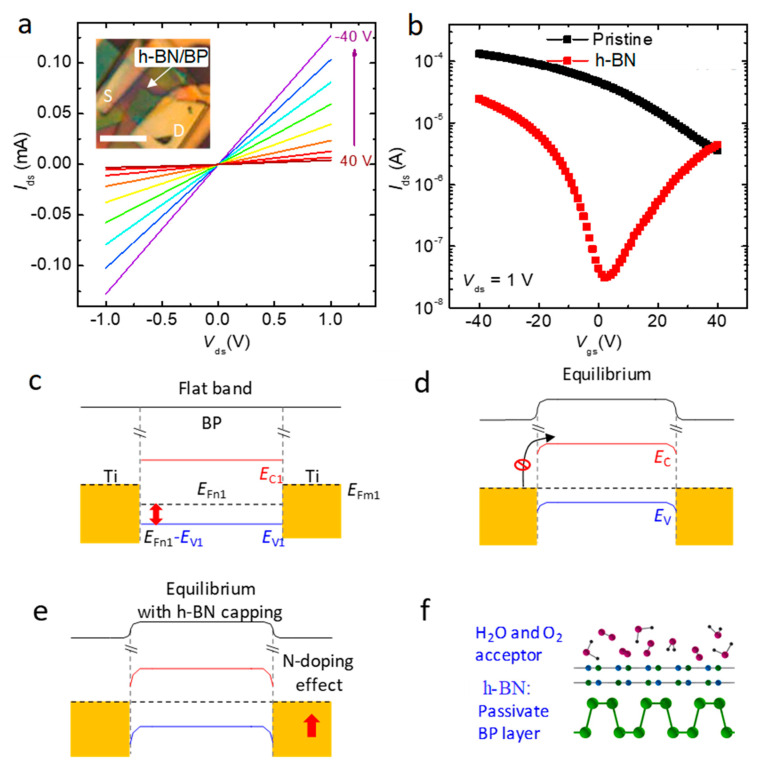
Electrical characteristics of h-BN-capped BP FET devices and mechanisms underlying the formation of ambipolar BP. (**a**) Output curves obtained from h-BN-capped devices, as a function of the gate bias. The inset shows an OM image of the device. The scale bar indicates 5 µm. (**b**) Transfer curves obtained from the pristine and h-BN-capped BP devices. (**c**,**d**) A band diagram of the pristine BP FET under flat band and equilibrium state. (**e**) The equilibrium state of the band diagram for h-BN-capped devices. (**f**) Schematic diagram describing the n-doping function of the h-BN capping layer on the BP layer. The h-BN protected the BP from acceptor adsorbates, such as water or oxygen, present in the ambient air.

**Figure 3 membranes-11-00952-f003:**
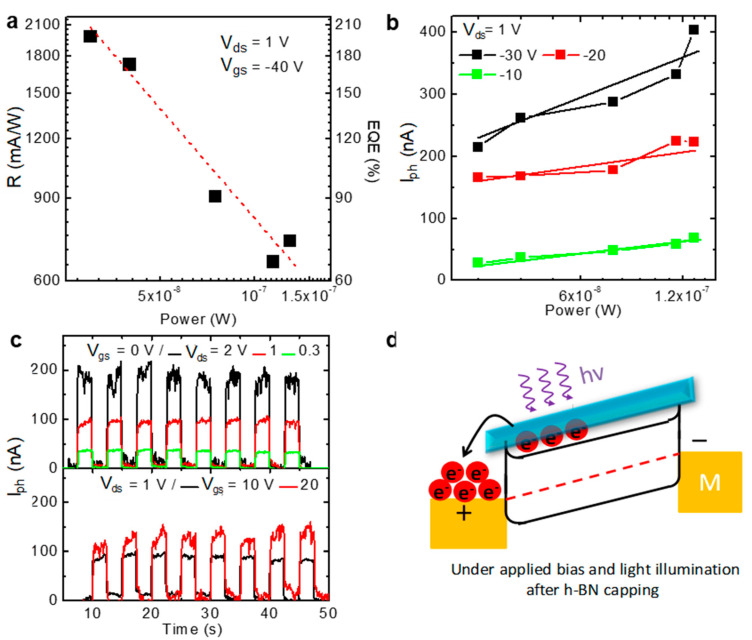
The application of the BP FETs in a photodetector. (**a**) The photoresponsivity of the BP device. (**b**) The photocurrent as a function of the incident illumination power, which shows a monotonic increase in the photocurrent. (**c**) The pulse photocurrent response dependent on the drain and gate bias up to 50 s. (**d**) The fast photocurrent response due to transparent h-BN capping effects on the barrier at the contact junction under bias application with light illumination.

**Table 1 membranes-11-00952-t001:** Comparison of figures of merits for photodetectors based on 2D materials.

Material	Measurement Conditions				*R* (mA/W)	Resp. Time (ms)	Spec-Tral Range	Reference
	*V*_ds_ (V)	*V*_gs_ (V)	*λ* (nm)	*P*				
>1L BP	1	−40	500	30 μW	10,000	<14	UV-Vis-IR	This work
		0			510			
		60			753			
>1L BP	0.05	0	633	60 nW	76	100	Vis-IR	Ref. [21]
>1L BP	0.02	0	640	10 nW	5	1	Vis-IR	Ref. [6]
1L MoS_2_	8	−70	561	150 pW	880 k	4000	Vis	Ref. [35]
1L MoS_2_	1	50	532	80 μW	8	50	Vis	Ref. [28]
>1L MoS_2_	1	−2	633	50 mW/cm^2^	110	>10k	Vis-IR	Ref. [36]
>1L WS_2_	1	-	458	2 mW	21m	5.3	Vis	Ref. [29]
>1L GaTe	5	0	532	30 uW/cm^2^	10m	6	Vis	Ref. [30]
>1L GaSe	5	0	254	1 mW/cm^2^	2800	300	UV-Vis	Ref. [31]
>1L GaS	2	0	254	256 uW/cm^2^	4200	30	UV-Vis	Ref. [32]

*P* is light power; *R* is responsivity; Resp. time means response time; UV is ultraviolet; Vis is visible; IR is infrared.

## Data Availability

Data are contained within the article.

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
