# Peer review of "High Photoresponse Black Phosphorus TFTs Capping with Transparent Hexagonal Boron Nitride"

_membranes, 2021, doi:10.3390/membranes11120952_

Round 1

Reviewer 1 Report

Despite the exciting topic for the readers of “membranes”, the manuscript has some serious flaws, which need to be addressed. I hope you can correct all the below-mentioned flaws in the frame time of a major revision. Find the comments in the attachment.

Author Response

we provided a point-by-point response in doc.

Reviewer 2 Report

The manuscript "High Photo Response Black Phosphorus TFTs Capping with Transparent Hexagonal Boron Nitride" presents the ambipolar characteristics of a few-layer BP capped with a transparent hexagonal boron nitride. The authors demonstrated p-type to n-type conversion, which is interesting in terms of high electron mobility devices. The topic of this work is appealing in the context of flexible and transparent 2D devices, and it has the potential to provide interesting electronic and optoelectronics features. The datasets are clearly presented, and the text is clear. I would recommend the publication of the manuscript after addressing the following points.

  1. Regarding the state-of-the-art, I strongly suggest the authors improve the introducing part and give a detailed discussion to the more recent previous works on this topic to put this work into the general context of the community. Regarding previous works on capping and protecting BP few-layers, there are many recent references that the authors should discuss better to show this work's novelty, such as:
    Doi: 10.1038/s41566-021-00787-x
    Doi: 10.1103/PhysRevB.103.L041407
    Doi: 10.1364/OME.391725
    Doi: 10.1021/acsanm.1c00351
    Doi: 10.1039/C6NR02554D
    Doi: 10.1039/D0TC00740D
  2. What was the authors' motivation for using hBN as a capping layer? Did the authors try other 2D insulating flakes instead of hBN?
  3. In Figure 1a, the boundary of the BP flake should be determined.
  4. English language should be improved and grammar checked. There are some writing mistakes in the text, such as in P5: blank Bracket ()?
  5. In p5, it is mentioned that "The electron mobility in the BP devices increased to 73 cm2/V.s, a record high electron mobility, to our knowledge". Please cite recent works that reported a lower value to compare to your results.

Author Response

(The authors gave the same response as above.)

Round 2

Reviewer 1 Report

Thanks to the authors for addressing the comments. I believe the quality of the paper now is acceptable for the journal of Membranes.